# Can audit and feedback improve health service readiness and delivery outcomes in a low-resource setting? Effectiveness results of the IDEAs strategy from central Mozambique

Aneth Dinis[1,2]*, Orvalho Augusto[2,3], Quinhas Fernandes[1], Ermyas Birru[2], Ruth Etzioni[4,5], Sarah Gimbel[2,6], Stephen Gloyd[2], Isaías Ramiro[7], Artur Gremu[7], Grace John-Stewart[2,8,9,10], Bradley H. Wagenaar[2,8], Bryan J. Weiner[2], Sérgio Chicumbe[11], Kenneth Sherr[2,8,12]

1 National Department of Public Health, Ministry of Health, Maputo City, Mozambique, 2 Department of Global Health, University of Washington, Seattle, Washington, United States of America, 3 Community Health Department, Eduardo Mondlane University, Maputo, Mozambique, 4 Department of Biostatistics, University of Washington, Seattle, Washington, United States of America, 5 Department of Health Systems and Population Health, University of Washington, Seattle, Washington, United States of America, 6 Department of Child, Family & Population Health Nursing, University of Washington, Seattle, Washington, United States of America, 7 Comité para Saúde de Moçambique, Maputo, Mozambique, 8 Department of Epidemiology, University of Washington, Seattle, Washington, United States of America, 9 Department of Medicine, University of Washington, Seattle, Washington, United States of America, 10 Department of Pediatrics, University of Washington, Seattle, Washington, United States of America, 11 National Institute of Health, Maputo, Mozambique, 12 Department of Industrial and Systems Engineering, University of Washington, Seattle, Washington, United States of America

* aneth@uw.edu

## Abstract

Limited evidence exists on audit and feedback(A&F) in low-resource contexts. We tested the Integrated District Evidence-to-Action (IDEAs), a multicomponent A&F strategy in Mozambique's maternal and child health (MCH) services. IDEAs include A&F meetings, readiness assessments, and facility support. We report the effectiveness results. IDEAs were implemented in 2016–2020 across two provinces, 12 districts, and 154 primary health facilities in Mozambique. We assessed 1) ten service delivery outcomes across antenatal, maternity, postpartum, childcare, and reproductive health services and 2) five service readiness outcomes (medicines, infrastructure, equipment, care, and staffing availability). We used propensity score matching to minimize bias and a controlled interrupted time series with a negative binomial mixed effects model for service delivery analysis, presenting incidence rate ratios (IRR) with 95% confidence intervals (95% CI). For service readiness outcomes, we created composite scores for each domain and a difference-in-difference analysis using an ordinal mixed effects model, reporting odds ratios (OR) and 95% CI. Significant associations were found with first at-risk child appointments (IRR = 1.06 [1.04, 1.07]), first polymerase chain reaction tests for HIV-exposed children (IRR = 1.02 [1.01, 1.03]), new contraceptive users (IRR = 0.95 [0.94, 0.96]), women starting

**Data availability statement:** The datasets for analyzing the impact on readiness outcomes have been uploaded to a public repository and can be assessed without restrictions: https://doi.org/10.7910/DVN/CVUIAX. The data for analyzing the impact on service delivery outcomes is non-public, governmental. The minimal dataset for this data can be sourced from the custodian of the data (www.misau.gov.mz) upon reasonable request.

**Funding:** The research reported in this publication is supported by the US National Institutes of Health (NIH) under award number 1R01HD092449-01A1 and the Doris Duke Charitable Foundation's African Health Initiative grant #2016106, both awarded to authors KS and QF. The authors AD, OA, QF, SGi, IR, AG, and KS received salary support from both grants. EB, GS, and BJW received salary support from NIH. The funders had no role in study design, data collection and analysis, the decision to publish, or the preparation of the manuscript.

**Competing interests:** The authors have declared that no competing interests exist.

long-lasting contraceptives (IRR = 0.94 [0.93, 0.95]), availability of infrastructure (OR = 5.84 [1.32, 25.88]) and essential care (OR = 0.13 [0.03, 0.54]). No significant associations were found between IDEAs and six of 10 service delivery outcomes (women with a fourth dose of preventive malaria treatment; women protected with tetanus vaccine; women with four+ antenatal visits; deliveries with active management of the third stage of labor; first postpartum consultations and fully vaccinated children) and on medicine, equipment, and staffing availability. We observed mixed effectiveness in implementing IDEAs, with null and sub-optimal effects suggesting the need to refine and adapt strategy components to more effectively address clinical and readiness outcomes.

## Introduction

Among the 17 Sustainable Development Goals (SDGs) led by the United Nations, the third Goal aims to end all preventable deaths under five years of age [1]. Neonatal deaths account for 47% of all under-five deaths worldwide [2]. While progress has been made, efforts are needed to achieve the SDG target of 12 or fewer neonatal deaths per 1000 live births by 2030 [2]. Survival rates for newborns vary greatly depending on the region, with sub-Saharan Africa having the highest neonatal mortality rate at 27 deaths per 1000 live births in 2020 [3]. A multitude of complex factors contributes to avertable neonatal deaths, including maternal characteristics, community-level determinants (such as access to healthcare, clean water, and proper nutrition during pregnancy and after birth), health systems factors (such as availability of resources, and quality of obstetric and pediatric care), and societal aspects associated with political stability and armed conflicts [4]. Notably, most causes of neonatal deaths are preventable and curable, including preterm and intrapartum complications, postpartum sepsis, and infectious diseases [5].

Clinical practice guidelines standardize treatment plans and help healthcare providers make evidence-based clinical decisions [6]. The potential benefits of guidelines include enhanced consistency in healthcare providers' practice, increased efficiency, and reduced morbidity and mortality. Despite these benefits, adherence to evidence-based guidelines is suboptimal [6,7] due to a lack of guideline awareness and content among providers, lack of time to fully apply guidelines, lack of resources and materials to implement guidelines, limited integration of guideline recommendations into organizational structures and processes, and provider resistance [7–9].

Audit and feedback (A&F) is an implementation strategy that summarizes clinical performance (audit) over a specific period and the provision of this summary information (feedback) to individual practitioners, teams, or health organizations [10]. A&F is frequently used to change health provider behavior and improve the quality of care [10]. A&F provides objective data on discrepancies between current practice and target performance, as well as comparisons of performance across health teams. By identifying performance gaps, it can influence action by encouraging providers and teams to minimize performance discrepancies [11]. Prior studies on the effectiveness

of A&F on clinical practice have shown variable (IQR of 0.5-16%), small to moderate effects in provider compliance to guidelines, without increasing impact over time [11–13]. Most A&F evaluations are from randomized control trials in high-income settings [12], and uncertainty remains about the potential benefits of A&F in low-income settings where the disease burden is higher, health systems are weaker, and baseline clinical practice guideline application has considerable space for improvement. Further evidence on the effects of A&F is needed from these settings.

In Mozambique, a country with limited resources and a high disease burden, neonatal mortality remains a public health problem, with a neonatal mortality rate of 28 deaths per 1,000 live births [14]. Utilization of maternal and child health (MCH) services in Mozambique is high, with an estimated 93% of pregnant women attending their first antenatal care (ANC) visits, 73% giving birth in a health facility, and 76% of children ages 12–23 months receiving the third pentavalent vaccination [15]. Despite this high utilization, the quality of evidence-based interventions delivered at or around the time of birth remains sub-optimal in Mozambique [16,17].

The Integrated District Evidence to Action (IDEAs) program tested a multi-component A&F implementation strategy that aimed to improve the coverage and quality of a bundle of existing evidence-based interventions targeting major causes of neonatal mortality. IDEAs hypothesized that iterative A&F cycles would directly improve routine use of clinical guidelines at the facility level while also enhancing health system readiness to support care provided by health providers, leading to improvements in the coverage and quality of service provision and reducing newborn mortality. In the context of IDEAs, *service readiness* means the availability of structural resources such as basic equipment, infrastructure, essential medicine, adequate care practices, and staff to provide maternal and child services; *service delivery* is the uptake of clinical maternal and child health services. This manuscript reports results from a quasi-experimental effectiveness evaluation of IDEAs in improving service readiness and service delivery outcomes targeting major causes of neonatal mortality in central Mozambique.

## Materials and methods

### Program description

The IDEAs strategy is designed to improve health service delivery using A&F tools that identify service gaps, address multi-level barriers and facilitators for delivery, and enable health workers to prioritize and test interventions to improve the application of Ministry of Health (MOH) guidelines for neonatal health. The IDEAs implementation strategy has been previously described elsewhere [18]. Briefly, IDEAs is an iterative three-step process of 1) health system readiness assessment; 2) A&F meetings to identify gaps and develop action plans (comprised of priority problems and *'micro-interventions'* defined as solutions identified, specified, and implemented at the health facility level); and 3) mentorship and funds to support action plan implementation (Fig 1). Meetings are held for each district, uniting facility MCH leads with district and provincial supervisors to test a scalable model that could integrate into the Mozambique health system. Implementation of the IDEAs strategy was led by district MCH managers, targeting MCH nurses at the facility level and supported by managers at the provincial level. Cycles are designed to be semiannual and were implemented between October 2016 and December 2020. IDEAs design incorporates best practices elements for designing A&F interventions as suggested by Ivers at al [11].

### Steps involved in the IDEAs implementation strategy (Fig 1).
*Step 1: Facility and district service readiness assessment.* Before each bi-annual five-day A&F meeting, the study team applied standardized service readiness assessment tools at three health facilities per district to assess structural readiness to deliver guideline-based care (such as staffing levels and availability of essential commodities, equipment, and supplies).

*Step 2: Audit and feedback meeting.* Facility, district, and provincial MCH managers participated in semiannual A&F meetings, auditing data from routine health information systems (RHIS) and service readiness assessments, and comparing performance relative to goals. Feedback is provided in graphical and tabular formats, and secular trends and performance measures are visualized. Each facility and district team presents its performance, followed by a group discussion to interpret results, identify barriers to guideline adherence, and develop action plans highlighting priority problems,

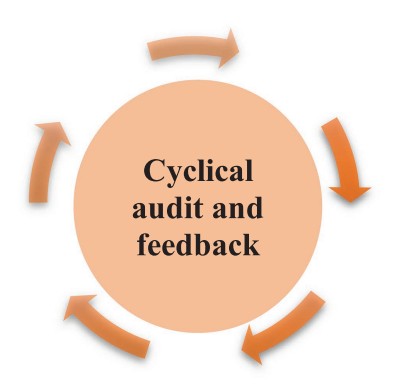

**IDEAs audit and feedback implementation strategy**

Manica and Sofala provinces in Mozambique
12 districts
154 health facilities

Led by maternal and child health managers

**Step 1: Facility and district readiness assessment**

⇒ Twice a year per 12 districts
⇒ Sample of 3 rotating facilities per district

**Cyclical audit and feedback**

**Step 2: Audit and feedback meetings**

⇒ Twice a year per 12 districts
⇒ Action plans developed or updated
⇒ Low and high performing facilities selected

**Step 3: Target facility support**

⇒ Action plans reviewed
⇒ Up to 2 supervision visits per cycle
⇒ Modest financial support

**Fig 1. Steps of the IDEAs audit and feedback strategy.**

specific and measurable targets, and resources required to implement micro-interventions. If the identified problem in the A&F meeting is, for example, "weak diagnosis of obstetric complications," specific micro-interventions pointed out by the MCH managers could be, for instance, evaluating all partograms every 15 days or conducting on-the-job training on managing the third stage of labor, etc.

Step 3: Targeted facility support. At each district A&F meeting, three facilities (one high-performing and two low-performing) were selected based on the performance assessment to receive up to two supervision visits per cycle from district and province supervisors. Modest monthly financial support (US$1,250) was provided to the districts to facilitate action plan implementation and supervision. During supervision visits, supervisors reviewed action plans, identified barriers to guideline implementation, and provided technical assistance to address context-specific barriers.

## Study design

The IDEAs strategy evaluation is guided by the Reach, Effectiveness, Adoption, Implementation, and Maintenance (RE-AIM) framework. All dimensions of the RE-AIM were used in planning, evaluating, and reporting IDEAs. The reach, adoption, implementation fidelity, and maintenance are reported elsewhere[18], and the present paper focuses on the effectiveness. We employed two separate analyses to evaluate the effectiveness of the IDEAs program.

To assess IDEAs impact on service delivery outcomes, we conducted a controlled interrupted time series analysis (ITS), while the impact on service readiness was evaluated through a difference-in-difference (DID) approach [19,20].

## Study setting

Intervention health facilities were located in Manica and Sofala provinces in central Mozambique, with a combined population of over 4 million inhabitants [21]. The IDEAs strategy was implemented in 154 primary-level health facilities across 12 districts, representing 63% and 43% of primary healthcare facilities in Manica and Sofala, respectively (Table 1). To maximize the potential impact of resource investment, intervention districts were selected based on their large population size, robust health facility network, and geographic accessibility. Districts were chosen as the intervention unit as they are the logical disseminating agents for delivering interventions to subordinate facilities, can access resources to meet health facility needs, and have the authority to implement management decisions within health facilities in their districts.

### Health facility sample.

Two facility samples were used to analyze the IDEAs intervention's impact on 1) service delivery outcomes and 2) service readiness (S1 Fig).

*Service delivery outcomes sample (S1A Fig)*: Service delivery outcomes sourced from the RHIS were compared between 154 intervention facilities and 349 control facilities from the neighbors Zambezia and Tete provinces. All intervention and control sites are facilities that offer primary health care services, which include rural and urban centers (93%), health posts (4%), and rural and district hospitals (3%). The inclusion criteria for control facilities included being a public sector health facility providing primary health care services and having data available on selected outcomes since October 2016.

*Service readiness sample (S1B Fig)*: Prior to implementation, 12 control districts were selected from Zambezia, Tete, and Sofala provinces, matched 1:1 to intervention districts based on district population size, facility network, and geographic attributes (e.g., urbanicity and distance from the provincial capitals). Within each district, three public sector health facilities were selected, including the highest volume facility in each district, plus two additional randomly selected units (totaling 36 intervention and 36 control facilities). All 36 intervention facilities are a subset of the 154 intervention and 33 of the 349 control facilities described above. Three additional facilities from one district of Sofala Province that are not included in the overall 349 control facilities were sampled to complete this subset.

## Data sources

Health facility monthly data on service delivery outcomes were extracted from the Mozambique MOH's RHIS (the *Sistema de Informação de Saúde para Monitoria e Avaliação – SIS-MA*) covering the period from October 2016 through December

**Table 1. IDEAs program setting.**

| Province | Population (2017)[a] | Total Districts (2021)[b] | Districts with IDEAs | Total Health facilities (2021)[b,c] | Health facilities with IDEAs |
|---|---|---|---|---|---|
| Manica | 1,911,237 | 12 | 7 | 126 | 79 (63%) |
| Sofala | 2,221,803 | 13 | 5 | 174 | 75 (43%) |
| Total | 4,133,040 | 25 | 12 | 300 | 154 |

[a]National population and housing census 2017;

[b]Provincial statistical data 2021;

[c]Excluding provincial hospitals.

2021. Service readiness data on the availability of medicines, infrastructure, essential care, equipment, and clinical staff were collected by project study teams through four annual health facility assessments across 72 health facilities (36 intervention and 36 control sites) using the World Health Organization Service Availability and Readiness Assessment (SARA) tool [22].

**Study variables.**

1) **Service delivery outcomes**

Multiple criteria guided the selection of service delivery outcome indicators, including 1) evidence of their theoretical relationship to neonatal mortality reduction [23–25], 2) being a focus of A&F during the IDEAs meetings, and 3) their availability in the RHIS. Monthly counts of ten indicators from antennal care, maternity, postpartum, child consultations, and family planning were extracted from the RHIS for each health facility, including the number of 1) pregnant women with a fourth dose of intermittent preventive treatment for malaria (IPTp); 2) pregnant women with a second-to-fifth dose of tetanus vaccine; 3) pregnant women with a fourth or more antenatal care visit; 4) deliveries with active management of the third stage of labor; 5) first postpartum consultations; 6) fully vaccinated children; 7) first at-risk child appointments; 8) first PCR tests for HIV-exposed children; 9) new users of contraceptive methods, and 10) women starting long-lasting methods of family planning. This last indicator was created by aggregating the counts of the three long-lasting contraceptive methods (intrauterine device, implant, and injectable Depo Provera).

2) **Service readiness outcomes**

We created composite measures for each of the following domains, using the list of items available in the SARA tool for each domain: Medicine availability, infrastructure availability, services provided, and equipment availability. The list of items used to create composite scores is detailed in supplementary information in S1 Appendix. In summary, fifteen items were included to create composite scores for essential medicines, which were selected based on the World Health Organization's list of priority life-saving medicines for women and children [26]. For infrastructure availability, six groups of items describing the availability of communications, power supply, basic amenities, processing of equipment for reuse, and infection control were included. Sixteen items were included to create composite scores for services provided, focusing on those delivered at or around the time of birth with a known influence on neonatal mortality. Twenty-eight items were used to create the composite score for the availability of essential equipment, which were also selected, given their relationship with neonatal mortality prevention and according to the SARA list of items for these categories (S1 Appendix) [22]. Counts were used to describe the availability of human resources, specifically the number of MCH nurses. We focused only on MCH nurses because they are the principal cadre of professionals providing MCH services in primary care in Mozambique and are directly linked with the IDEAS strategy.

Composite scores for each health facility were estimated by dividing the observed available items by the total number of possible items for each category. For instance, the total possible number of items for the essential medicines domain is 15; for a health facility with eight items available, the medicine score for that facility would be 8/15 = 0.53. The resulting composite scores are discrete and inappropriate to treat as continuous variables; therefore, we preserved the ordinal nature of the score using statistical techniques for ordinal outcomes.

3) **Explanatory variables**

For our analysis of service delivery outcomes, we included the following covariates: Facility location (rural or urban), type of facility (rural health center, urban health center, rural hospital, district hospital), provincial per capita GDP, Covid-19 pandemic period (as a binary variable of prior to or after Covid-19 mitigation policies were put in place - cutoff in March of 2020), and MOH catchment area size. Health facility location and the count of maternal and child nurses were included in our analysis of service readiness outcomes.

**Statistical analysis.** *First analysis: Impact on service delivery outcomes*

We applied a controlled interrupted time series (ITS) analysis covering October 2016 through December 2021 to estimate the impact of the IDEAs program on service delivery outcomes. Data on study outcomes prior to October 2016 were not included due to the MOH's adoption of a new RHIS based on the DHIS2 (District Health Information Software 2) [27] and the resulting data inconsistencies.

A total of nine IDEAs cycles were completed during the intervention period. Because of the complexity of the intervention and its iterative nature, it is unlikely that the IDEAs strategy led to observable improvements during the first two cycles. Therefore, we parametrized 63 months of available data in the time series analysis as follows:

- October 2016 through August 2017 (11 months) as the pre-intervention period,

- September 2017 through December 2020 (40 months) as the intervention period,

- January 2021 through December 2021(12 months) as the post-intervention period.

Therefore, the ITS model for service delivery analysis has three segments of changes in slopes and no immediate change segments – see the formula below:

$$
\begin{aligned}
Y_{jt} = {} & \beta_0 + \gamma X_{jt} + \beta_1 time.pre\_intervention_{jt} + \beta_2 time.intervention_{jt} + \beta_3 time.post\_intervention_{jt} + \beta_4 treatment_{jt} \\
& + \beta_5 pre\_intervention * treatment_{jt} + \beta_6 intervention * treatment_{jt} + \beta_7 post\_intervention * treatment_{jt} \\
& + \beta_8 \sin \frac{2\pi.time}{12} + \beta_9 \cos \frac{2\pi.time}{12}
\end{aligned}
$$

Where:
j = facility; t= time
$\beta_0$ = Baseline level of the outcome at time = 0
$\beta_1$ =A continuous variable indicating time in months from the start of the observation period is the mean change each month before the intervention.
$\beta_2$ = Is a continuous variable counting the number of months during the intervention (is 0 before), is the change in slope comparing intervention to pre-intervention
$\beta_3$ = Is a continuous variable counting months after the intervention (is 0 before), is the change in slope comparing the post-intervention to the intervention period.
$\beta_4$ = Is a variable indexing treatment (1) versus control (0), measures the difference in the outcome level prior to intervention, comparing treatment to control.
$\beta_5$ = Measures the difference in slope between treatment and control sites before intervention.
$\beta_6$ = Measures the difference in slope between treatment and control during the intervention period, which is the effect of the intervention.
$\beta_7$ = Measures the difference in slope between treatment and control in slope after the intervention period.
$\beta_8$ and $\beta_9$ are sine and cosine for capturing seasonality.
$\gamma$ is the vector of coefficients, and X is a matrix of covariates

To mitigate baseline confounding due to non-randomization, we conducted a 1:2 propensity score matching to balance the characteristics of intervention and control facilities [28]. Health facilities were matched based on geographic location, type of health facility, distance from reference facility, distance from provincial health services, number of MCH nurses, total beds, and health facility catchment area. We assessed the balance of the control and intervention groups by visualizing the distribution of propensity scores and comparing descriptive statistics. After matching, 150 intervention facilities and 300 control facilities were included in the analysis (S1 Table).

A truncated negative binomial generalized linear mixed effects model using template model builder (TMB) implemented in the glmmTMB library was applied to model the data of nine service delivery outcomes given that they are counts and do

not include zeros [29]; the exception is for the counts of women starting long-lasting methods of family planning, in which we fit a standard negative binomial. A random effect for the matched pair of facilities and the covariates (type of facility, location, COVID period, health facility catchment area size, and province GDP) was added to the model. We report the exponentiated coefficients as incidence rate ratios (IRR) and their 95% confidence intervals.

Sensitivity analysis was conducted by removing observations beyond six times the standard deviation of the outcome means in the matched data. The results did not meaningfully change the direction, magnitude of coefficients, or the conclusions.

*Second analysis: Impact on service readiness*

We applied difference-in-difference analysis to assess the impact of the IDEAs program on service readiness. Four time points of district readiness assessments were available, with an irregular frequency between them. We used the first assessment as the baseline and the average of three subsequent assessments as the intervention period, expressed in the following formula:

$$Y_{jt} = \beta_0 + \gamma X_{jt} + \beta_1 time_{jt} + \beta_2 treatment_{jt} + \beta_3 time_{jt} * treatment_{jt} + u_{jt} + \varepsilon_{jt}$$

Where:

j = health facility; t= time

$\beta_0$ = Baseline level of the outcome at time =0

$\beta_1$ = Is 1 after baseline and 0 before, measures the relative change in the control group.

$\beta_2$ = 1 for treatment and 0 for control, measures the odds in treatment compared to the control group at baseline.

$\beta_3$ = Is the effect of DID. It measures the change in odds of the treatment group above and beyond the change in odds in the control group.

$\gamma$ is the vector of coefficients, and X is a matrix of covariates

$u_{jt}$ = Is the random effect for the matched pair of facilities

$\varepsilon_{jt}$ = Error term

Cumulative link mixed models from the ordinal package were used to account for the ordinal nature of the composite scores for the following readiness domains: medicine, equipment, infrastructure, and essential care [30,31]. For the staffing domain, we use a Poisson model. A random effect for the facility and covariates previously described was added to the models. We present odds ratios (OR) and 95% CI for composite readiness score outcomes and IRR and 95% CI for staffing. All analyses were performed in R version 4.2.2.

**Ethics approval.** All the research methods were performed following the relevant guidelines and regulations. The study was approved by the institutional review board of the University of Washington (IRB#STUDY00003926), Mozambique's National Bioethics Committee for Health (CNBS-IRB00002657), and the Ministry of Health after endorsement from Manica and Sofala Provincial Health Directorates. Participant consent was not required for this analysis.

## Results

S1 Table describes the distribution of covariates in facilities in unmatched and matched data at baseline (October 2016). Most facilities are rural health centers located, on average, more than 50 kilometers from the referral health facility. Intervention facilities tend to have a slightly higher number of MCH nurses, include more health posts, and are more likely to be located in an urban center than controls.

### Results of the first analysis: impact on service delivery outcomes

Fig 2 presents the IRRs for time slopes from the adjusted analyses. In the next paragraph, we describe the intervention's effect, measured by the interaction between intervention time and treatment. Detailed results of the unadjusted and adjusted analysis are presented in supplementary S2 Appendix.

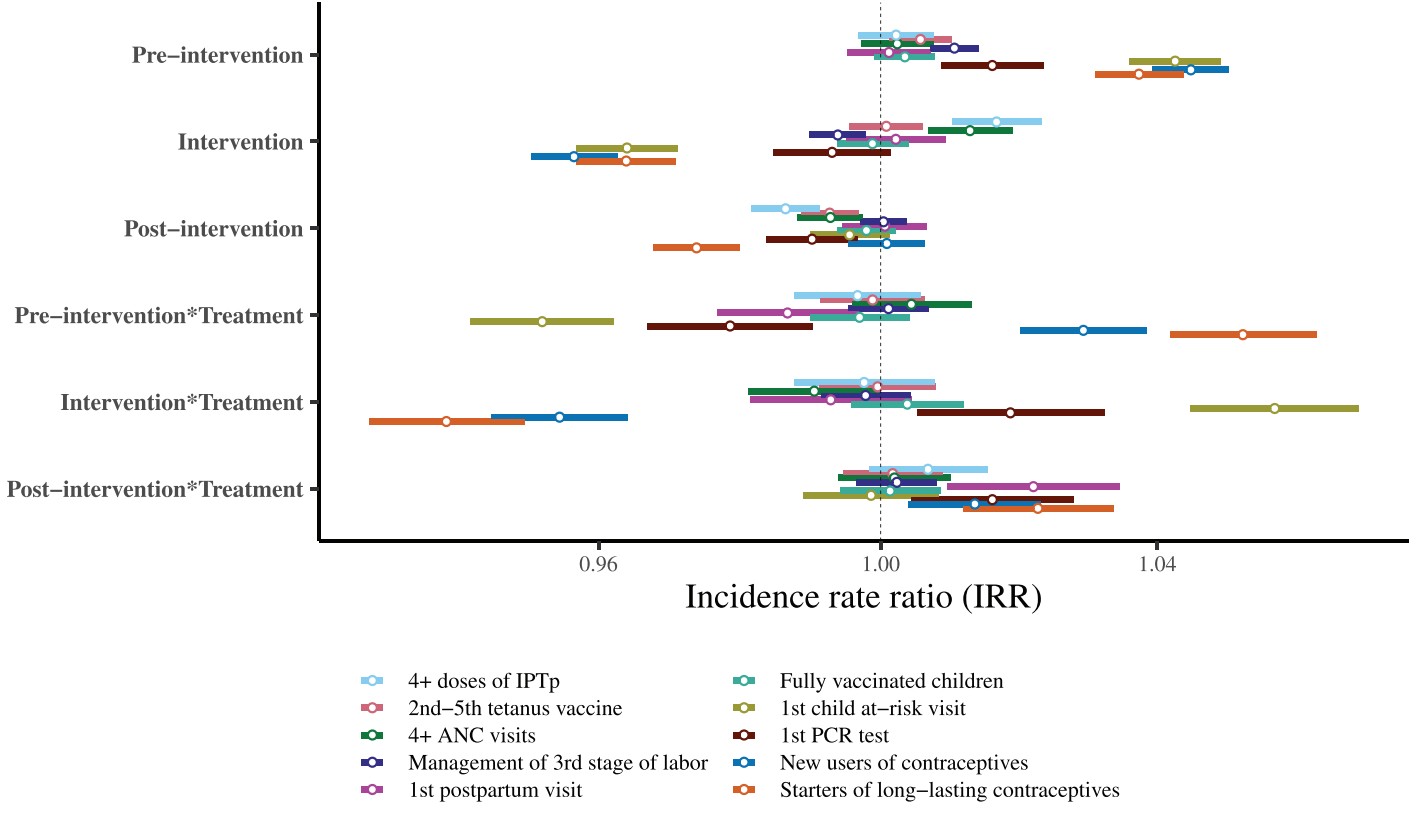

Fig 2. Incidence rate ratios (IRRs) of the slopes of adjusted models for service delivery outcome analyses. Note Fig 2: Adjusted monthly changes in slopes of service delivery outcome. The interaction **Intervention \*Treatment** measures the impact of the intervention. IPTp: Intermittent preventive treatment for malaria; PCR: Polymerase chain reaction.

We observed significant positive associations for two outcomes: The first at-risk child consultation (IRR = 1.06 [95% CI, 1.04, 1.07]) and the first PCR testing for exposed HIV children (IRR = 1.02 [95% CI, 1.01, 1.03]). In summary, we observed a monthly increase of six percentage points in the counts of first at-risk child consultations and two percentage points in the first PCR test performed in intervention facilities compared with control facilities.

Our analysis found non-significant associations between IDEAs and six service delivery outcomes: The fourth dose of preventive treatment for malaria (IRR = 1.00 [95% CI, 0.99, 1.01]); the second-to-fifth dose of tetanus vaccine (IRR = 1.00 [95% CI, 0.99, 1.01]); the fourth antenatal care visit (IRR = 0.99 [95% CI, 0.98, 1.01]); active management of the third stage of labor (IRR = 1.00 [95% CI, 0.99, 1.00]); the first postpartum consultation (IRR = 0.99[95% CI, 0.98, 1.00]); and fully vaccinated children (IRR = 1.00 [95% CI, 0.99, 1.01]).

Our analysis found significant negative associations for two outcomes: New users of contraceptives (IRR = 0.95 [95% CI, 0.94, 0.96]) and women initiating long-lasting contraceptives (IRR = 0.94 [95% CI, 0.93, 0.95]), respectively. In summary, we observed a monthly reduction of five percentage points in the counts of new users of contraceptives and six percentage points in the initiation of long-lasting contraceptives in intervention facilities compared with control facilities.

Fig 3 presents time series plots of observed data (transparent points), model trends (solid lines), and 95% confidence intervals (dashed lines) for all ten service delivery outcomes in intervention and control facilities as rates per thousand people.

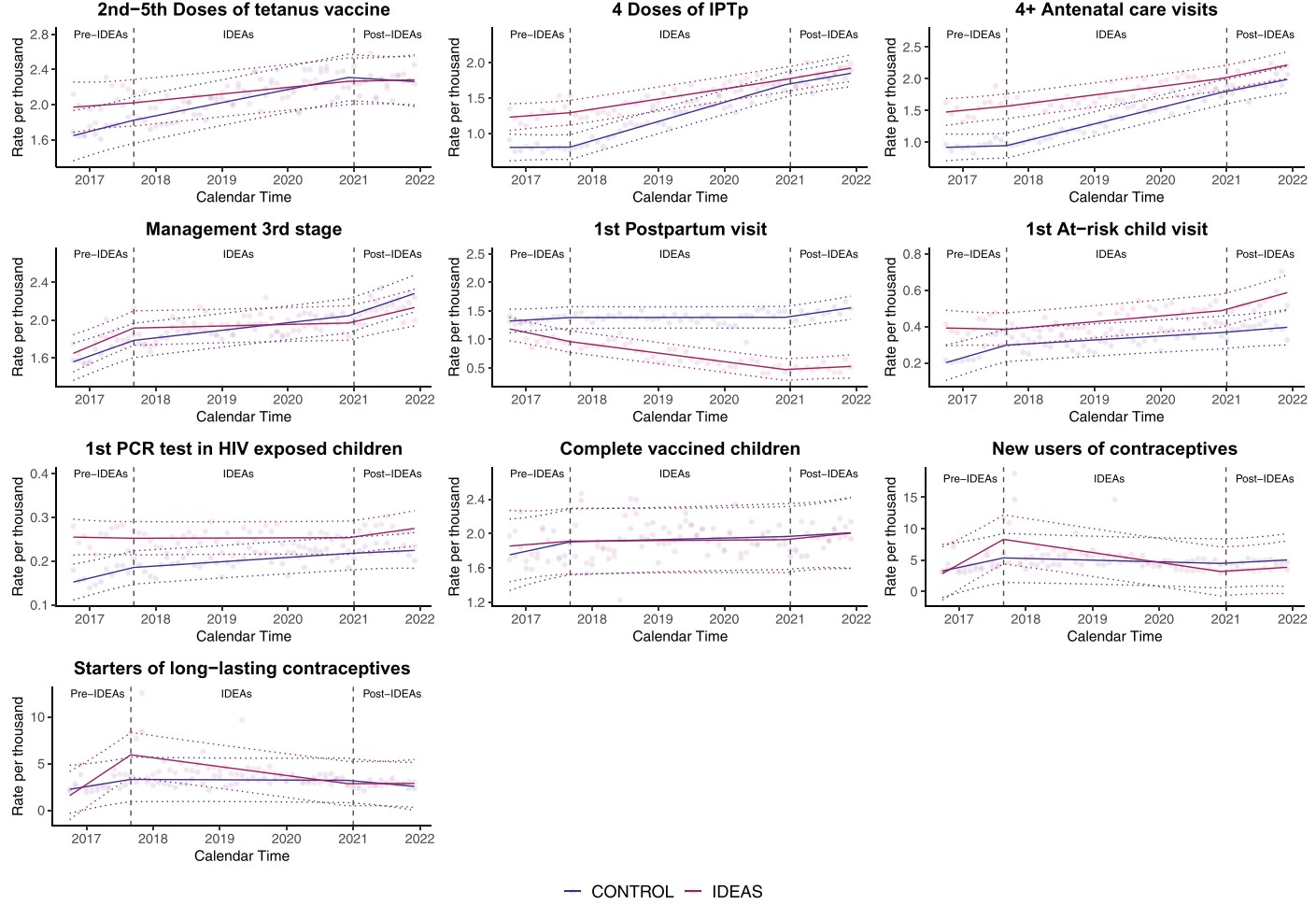

**Fig 3. Time series plots for ten service delivery outcomes in intervention (IDEAs) and control facilities.** Note Fig 3: IPTp: intermittent preventive treatment for malaria.

### Results of the second analysis: impact on service readiness outcomes

Our analysis on IDEAs impact on service readiness outcomes found that the odds for better infrastructure availability were 5.84 times greater in intervention facilities compared with control facilities (OR = 5.84 [95% CI, 1.32, 25.88]), whereas the odds of better availability of essential care services were 0.87 times less likely in intervention facilities compared with control facilities (OR = 0.13 [95% CI, 0.03, 0.54]) (Table 2). Null effects were found on the availability of essential medicines (OR = 2.61 [95% CI, 0.66,10.29]), essential equipment (OR = 1.20 [95% CI, 0.31, 4.64], and MCH staffing (IRR = 1.09 [95% CI, 0.82, 1.45].

## Discussion

We discuss a quasi-experimental evaluation of the IDEAs A&F strategy, an implementation approach led by MCH managers in 12 districts of Mozambique. The strategy aimed to improve adherence to MOH guidelines addressing key causes of neonatal mortality and enhance the health system's ability to implement these guidelines.

Our analysis found mixed results on IDEAs impact on service delivery and service readiness outcomes. In the analysis of IDEAs association with service delivery outcomes, the intervention was associated with improvements in at-risk

**Table 2. Results of the difference-in-difference analysis for service readiness outcomes.**

| Model Coefficients | Outcomes | | | | |
|---|---|---|---|---|---|
| | Medicine | Infrastructure | Equipment | Essential care | MCH-Staffing |
| | OR [95% CI] | | | | IRR [95% CI] |
| Mean change in controls | 0.30 [0.11, 0.83] | 0.26 [0.09, 0.73] | 0.95 [0.36, 2.49] | 8.23 [2.83, 23.97] | 0.85 [0.69, 1.05] |
| Difference at baseline | 0.99 [0.31, 3.19] | 0.32 [0.08, 1.22] | 0.85 [0.25, 2.86] | 4.11 [1.43, 11.81] | 0.82 [0.55, 1.21] |
| Effect DID | 2.61 [0.66,10.29] | **5.84 [1.32, 25.88]*** | 1.20 [0.31, 4.64] | **0.13 [0.03, 0.54]*** | 1.09 [0.82, 1.45] |
| Location (Urban) | 0.14 [0.03, 0.66] | 0.32 [0.06, 1.76] | 0.61 [0.13, 2.96] | 0.26 [0.08, 0.84] | 2.71 [1.83, 4.00] |
| MCH Staffing | 1.32 [1.19, 1.47] | 1.30 [1.17, 1.45] | 1.34 [1.20, 1.49] | 1.14 [1.06, 1.22] | -------------------- |

DID: difference-in-difference; *statistically significant at 0.05 level

child care outcomes, was not associated with changes in antenatal care, maternity, postpartum care, or well-child service outcomes, and was negatively associated with family planning outcomes. In our analysis of service readiness outcomes, IDEAs were associated with improved infrastructure availability, a reduction in the availability of essential care practices, and were not associated with changes in the availability of essential medicine, essential equipment, or staffing.

The observed association between IDEAs and improvements in at-risk child outcomes may highlight a more solid availability of resources in the verticalized HIV services compared with other services, indicating inequity in resource distribution within the health facilities. The observed null effects across most service delivery and readiness outcomes may indicate the complexity around implementing the intervention or that the strategy did not directly influence changes in those indicators, which may be linked to poor alignment and/or implementation of planned micro-interventions due to a lack of resources and inefficient oversight of these actions. Though the IDEAs strategy included the provision of moderate, flexible funding to districts to support the implementation of facility-level micro-interventions, these funds may have targeted infrastructural improvements (including items such as communications, transportation, power supply, and infection control infrastructure) instead of funding the purchase of essential medicines or hire additional staff (which are typically procured at national or provincial levels and is beyond the scope of districts' procurement capacity). Additionally, the modest funding provided by IDEAs may have been insufficient to move the needle on service readiness needs in an underfunded health system with high disease burden and patient loads.

We observed associations between the intervention and attenuated improvements in the availability of essential care practices for pregnant women and newborns and family planning indicators. Reasons for sub-optimal results in essential care scores might include a lack of resources (which could explain the lack of improvements in the availability of essential medicines, equipment, and human resources) and the need to reinforce clinical supervision and strengthen clinical training in obstetric care management. This last point is reinforced by the process evaluation of the IDEAs in which "*weak diagnosis and management of obstetric complications*" was consistently identified by the MCH nurses as a priority problem in IDEAs A&F meetings [18].

The observed reduction in family planning indicators across both intervention and control facilities may be associated with external factors that affected these services in many low-income countries during the intervention period. As an example, the United States 'Global Gag Rule' which prohibits non-US-based non-governmental organizations (NGOs) from providing, referring for, or counseling on abortion as a method of family planning, did fracture the provision of sexual and reproductive health by reducing contraceptive service availability, accessibility and providers training in LMICs [32–34]. Additionally, multiple studies have reported negative trends in providing family planning services in LMICs during the COVID-19 pandemic, including Mozambique [35,36]. Indeed, our analysis demonstrates that there was a monthly reduction of nine percentage points in new contraceptive users and a three percentage point reduction in initiation of long-lasting contraceptives in IDEAs intervention facilities during the COVID period (S2 Appendix). Although these

external factors are real, it is unclear why the decrease in observed family planning delivery indicators was greater in intervention than in control facilities.

Similar to our study, existing evidence on the impact of A&F on clinical outcomes shows mixed results. A Cochrane review of 49 A&F studies found a risk difference of -0.4% (IQR -1.3% to 1.6%) decrease in patient outcomes (for studies with dichotomous outcomes) and 17% (IQR 1.5% to 17%) improvement in patient outcomes (for studies with continuous outcomes) [13]. However, most of these studies are randomized controlled trials from high-income countries [13,37,38]. An overview of systematic reviews of strategies to improve the quality of maternal and child healthcare in low and middle-income countries included ten reviews on A&F, in which nine detected small to moderate effects on professional practice, and one revealed ineffectiveness with a magnitude of effects ranging from -17% to 49% change. For impact on health outcomes, no statistically significant differences were found in one review included in the overview [39]. Another review stated that complex areas such as disease management, adherence to guidelines, and diagnosis appear less affected by A&F; in contrast, prescription and preventive care activities are more likely to change with A&F strategy due to differences in the complexity of the levels of decision-making for providers in these areas of care [40].

This evaluation has some limitations. First, we recognize that the complexity of the intervention and the number of indicators pose challenges to the evaluation. To assess IDEAs effects on service delivery outcomes, our evaluation relied on aggregate monthly facility-level count data that are not ideally suited to assessing improvements in the application of clinical guidelines, nor were we able to measure changes in neonatal mortality. Ideally, our assessment of service delivery outcomes would include more granular assessments of the application of MOH guidelines at the facility level. However, in the absence of these data (and given resource constraints precluding setting up such a system – like an electronic health record system – for this study), we relied on RHIS data that we acknowledge has limitations. Data on selected outcomes before October 2016 were unavailable in the system, and as a result, our time series began concurrently with the initiation of the intervention. We expect the effects of this to be minimal given that the periodicity of the IDEAs intervention was semiannual, and if there were any effects, they would most likely be conservative, biasing results towards the null. We were also unable to assess the intervention's impact on neonatal mortality due to the lack of reliable facility-level mortality data or community survey data during the analysis period. Nevertheless, routine health information systems have strengths, such as their availability and representativeness throughout the Mozambique public sector health network and potential for improvement [41].

In addition, residual confounding may exist. Since the propensity score matching was not perfect, we further adjusted our models with covariates to minimize the remaining source of confounding bias. Our analysis of service readiness outcomes relied on only four data points (four annual cross-sectional readiness surveys), which may have reduced our power to detect changes in service readiness associated with the intervention. Furthermore, our service readiness measures are composite outcomes, which may obscure improvements in individual components of the aggregate outcome. Finally, given the quasi-experimental design of this study, care should be taken in attributing causality of observed results to the study intervention.

Despite these limitations, this study has important strengths. The intervention was delivered by district health teams and implemented in 12 districts and over 150 health facilities, providing relevant results to guide further scale-up of the intervention in Mozambique. The quasi-experimental design that includes appropriate controls and robust analytic techniques addresses multiple sources of bias. Additionally, most service delivery outcomes included in the study were linked to the principal issues identified by district MCH managers during A&F meetings, and assessing improvements in these targeted indicators is appropriate for evaluating the IDEAs intervention. Finally, we have evaluated a strategy implemented in a pragmatic, real-world practice setting in a resource-limited environment, narrowing the evidence gap about the effectiveness of A&F processes outside of high-income contexts.

## Recommendations and future steps

The analysis of the IDEAs implementation process showed that the strategy allowed MCH managers to identify problems and propose micro-interventions at the primary health facility level[18]. However, implementing micro-interventions

identified in action plans requires adequate resource availability, clinical supervision tailored to the identified problems and solutions, and close monitoring of proposed activities. Although the IDEAs design includes modest district funding and selective health facility supervision to support action plan implementation, our results suggest that the strategy needs to be refined and adapted to maximize improvements in service delivery and readiness outcomes. We recommend considering allocating funding or required resources at the level where the problems are identified (health facility level) to facilitate using those resources to solve context-specific challenges. This approach will require administrative support to health facilities to manage resources effectively.

The supervision component must also be tailored to reinforce clinical practice with a strong focus on managing obstetric care, which likely implies including highly skilled professionals in supervision teams.

Understanding how intervention fidelity influences its effectiveness is crucial to detecting which components of the IDEA's strategy (and other A&F interventions) are core and which elements can and should be adapted to enhance contextual appropriateness. Moreover, a deep understanding of barriers and facilitators for action plan implementation from the perspective of health system managers at the facility, district, and provincial levels will complement the evidence of this evaluation. Lastly, testing different combinations of A&F with other strategies – for instance, A&F and training of MCH nurses in the management of emergency obstetric care - and the relative impact of these combinations on clinical and health system readiness outcomes is needed across resource-limited settings to build the evidence base on this promising system-focused strategy.

## Conclusion

The IDEA's audit and feedback strategy presented mixed results. The strategy supported MCH managers in identifying priority issues and proposing local solutions, and it was associated with improvements in selected indicators from child-at-risk services and infrastructure availability. Suboptimal and null effects across most indicators suggest that various health system factors increase the complexity of implementing the intervention in this context, indicating the need for adjustments in the strategy design, implementation and evaluation to maximize the impact. Therefore, we recommend refining and adapting IDEAs strategy components to target more directly clinical and readiness outcomes.

## Supporting information

**S1 Fig. Figure of sampling of health facilities included in the analysis.** The figure is divided into part A, containing the sampling of districts and health facilities used to assess service delivery analysis, and part B, with the subsampling for service readiness outcomes.
(TIFF)

**S1 Appendix. List of items used to create composite scores for service readiness outcomes.** It presents four different lists of items selected from the service availability and readiness survey to create composite scores for readiness outcomes: the availability of essential medicine, essential infrastructure, essential services provided, and essential equipment.
(DOCX)

**S1 Table. Table of baseline characteristics of study facilities in unmatched and matched data.** This table presents the characteristics of the baseline unmatched and matched data via propensity score. The variables displayed include the number of maternal and child nurses, the number of facility beds, type of facility, location, distance from the health department, distance from referral health facility, and population catchment area size.
(DOCX)

**S2 Appendix. Detailed results of service delivery outcomes analysis.** This Excel sheet displays the results of the unadjusted and adjusted statistical analysis for all ten service delivery outcomes.
(XLSX)

## Acknowledgments

We want to thank the Ministry of Health of Mozambique, Manica and Sofala provincial health departments, district MCH managers, and facility MCH nurses for collaborating in designing, implementing, and evaluating the IDEAs strategy.

## Author contributions

**Conceptualization:** Aneth Dinis, Quinhas Fernandes, Sarah Gimbel, Stephen Gloyd, Grace John-Stewart, Bradley H Wagenaar, Bryan J Weiner, Sérgio Chicumbe, Kenneth Sherr.

**Data curation:** Isaías Ramiro, Artur Gremu.

**Formal analysis:** Aneth Dinis.

**Funding acquisition:** Quinhas Fernandes, Kenneth Sherr.

**Methodology:** Aneth Dinis, Orvalho Augusto, Ermyas Birru, Ruth Etzioni, Sérgio Chicumbe, Kenneth Sherr.

**Project administration:** Isaías Ramiro.

**Supervision:** Quinhas Fernandes.

**Validation:** Orvalho Augusto, Kenneth Sherr.

**Writing – original draft:** Aneth Dinis.

**Writing – review & editing:** Aneth Dinis, Orvalho Augusto, Quinhas Fernandes, Ermyas Birru, Ruth Etzioni, Sarah Gimbel, Stephen Gloyd, Isaías Ramiro, Artur Gremu, Grace John-Stewart, Bradley H Wagenaar, Bryan J Weiner, Sérgio Chicumbe, Kenneth Sherr.

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
