## [Decision Letter · Decision Letter 0]

11 Dec 2024

PGPH-D-24-02343

Can audit and feedback improve health service readiness and delivery outcomes in a low-resource setting? Effectiveness results of the ideas strategy from central Mozambique

Dear Dr. Dinis,

Thank you for submitting your manuscript to PLOS Global Public Health. After careful consideration, we feel that it has merit but does not fully meet PLOS Global Public Health’s publication criteria as it currently stands. Therefore, we invite you to submit a revised version of the manuscript that addresses the points raised during the review process.

We look forward to receiving your revised manuscript.

Kind regards,

Guillaume Fontaine, PhD, RN

Academic Editor

Journal Requirements:

**Please only choose the relevant sentences from below**

1. Please clarify all sources of funding (financial or material support) for your study. List the grants (with grant number) or organizations (with url) that supported your study, including funding received from your institution. 

2. State the initials, alongside each funding source, of each author to receive each grant.

3. State what role the funders took in the study. If the funders had no role in your study, please state: “The funders had no role in study design, data collection and analysis, decision to publish, or preparation of the manuscript.”

4. If any authors received a salary from any of your funders, please state which authors and which funders.

2. In the online submission form, you indicated that "The data supporting this study's findings are available upon reasonable request from the corresponding author and with permission of the Ministry of Health of Mozambique and Manica and Sofala provincial health directorates.". 

3. Uploaded as supplementary information.

3. Please provide an Author Summary. This should appear in your manuscript between the Abstract (if applicable) and the Introduction, and should be 150–200 words long. The aim should be to make your findings accessible to a wide audience that includes both scientists and non-scientists. Sample summaries can be found on our website under Submission Guidelines:

https://journals.plos.org/globalpublichealth/s/submission-guidelines#loc-parts-of-a-submission

Additional Editor Comments (if provided):

Reviewers' comments:

Reviewer's Responses to Questions

**Comments to the Author**

1. Does this manuscript meet PLOS Global Public Health’s publication criteria ? Is the manuscript technically sound, and do the data support the conclusions? The manuscript must describe methodologically and ethically rigorous research with conclusions that are appropriately drawn based on the data presented.

Reviewer #1: Yes

Reviewer #2: Yes

2. Has the statistical analysis been performed appropriately and rigorously?

Reviewer #1: I don't know

Reviewer #2: I don't know

3. Have the authors made all data underlying the findings in their manuscript fully available (please refer to the Data Availability Statement at the start of the manuscript PDF file)?

Reviewer #1: Yes

Reviewer #2: Yes

4. Is the manuscript presented in an intelligible fashion and written in standard English?

Reviewer #1: Yes

Reviewer #2: Yes

5. Review Comments to the Author

Reviewer #1: General points

This is an impressive pragmatic intervention study to evaluate the impact of an audit and feedback intervention on maternal and child health service provision in Mozambique.

I think some delineation between the two studies is needed – especially in the methods and results sections. I found it a little hard to follow at times which study (readiness focused or outcomes focused) which subsection was referring to. It may be that a simple labeling of Study 1 and Study 2 are needed throughout the paper.

Also, I wonder conceptually about ‘service readiness’ precluding the delivery of an intervention. Should one assess service readiness before deciding to implement something. This could be done as a ‘screening’ process to identify needs before implementation a program. If a service is not ‘ready’ then is it reasonable to think that we can make significant changes and to maintain over time. Plus, it allows the opportunity to match gaps in readiness with strategies to support implementation. I wonder whether the authors had any thoughts about this and if so, could perhaps integrate this into the discussion.

Specific points

L49 – ‘a multi-component audit and feedback strategy’ - Perhaps a sentence summarizing the components within the intervention

L57 - Incidence rate ratios?

L61 – Define PCR?

L69 – ‘IDEAs were associated with improvements in selected indicators from child-at-risk services and available infrastructure’ - This seems selective given you report multiple positive associations in the results section. Perhaps drop or as in 'such as child-at-risk services’ etc.

L88 – state that SDG program is led by the United Nations.

L138 - Does ‘service readiness’ as a term perhaps need to be defined here or beforehand in the intro.

L144 – ‘address multi-level barriers and facilitators for delivery’ - How was this in particular operationalized? No real mention of barriers and enablers assessment in the paper. Is it referring to the readiness assessment? More clarity needed.

L162 – ‘Step 2: Audit and feedback meeting’ - You cite the Ivers 'no more business as usual' A&F paper which suggests several recommendations for developing A&F interventions. I wonder if it might be informative to the reader to map your A&F intervention to some of the features reported in the Ivers paper. I think this could speak to the point around the 'refinement' of the intervention which you mention in the discussion. So, which bits of the A&F intervention worked well, where are their opportunities to enhance the reports, can we build more theory into the A&F intervention etc.

L169 – what do you mean by micro-interventions?

L170 – ‘Step 3: Targeted facility support.” – Out of interest, are there are data on how these selected facilities fared in terms of outcomes? How did they compare to those that did not receive additional support. Did their selection lead to marked improvements in the low performing facilities?

L212 – ‘(totaling 36 intervention and 36 control facilities)’ - Are these facilities a subset of the 154 intervention and 349 control facilities? If so, would make this clear.

L228 – use acronym A&F

L365 - How were the exploratory variables selected? What was the criteria for judging their relevance.

L404 – Opening sentence of discussion too long, would suggest rewriting.

L417 – Not sure what ‘robust vertical resources’ means, perhaps define.

L421 – again more clarity about the micro-interventions needed

L458 - Mentioned that this study was an overview of systematic reviews.

L528 – ‘refining and adapting strategy components to target clinical and readiness outcomes more directly’ – really like this point but could you posit how you might approach this.

Reviewer #2: OVERALL

This is a well-written paper concerning the use of a recognised healthcare quality improvement technique applied in a new context. Unfortunately, my understanding of statistics is not sophisticated, so I cannot comment extensively on the statistical methods used. However, the presentation of incidence ratios for service delivery outcomes and odds ratios for service readiness outcomes seems reasonable as both are relatively straightforward to interpret for a range of skill levels.

TITLE

The Title adequately indicates the topic addressed and setting for this study.

ABSTRACT

The Abstract provides a good summary of the article.

INTRODUCTION

The Introduction is very informative about the context of the study and clearly emphasises the importance of finding ways to combat the major causes of neonatal mortality in central Mozambique.

METHODS

The Methods are clearly described and in a good level of detail. The details contained in Appendices/Supporting Information is very helpful.

RESULTS

The Results are well-presented, with a good balance between narrative and graphical information.

DISCUSSION

The Discussion is well-written and reflects the aims and findings of the study. The authors clearly identify and acknowledge the strengths of the paper and offer honest explanations of any weaknesses. Their conclusions are reasonable and well-expressed. The authors have included some recommendations within the Discussion section – it may be helpful for these recommendations to be in a separate section.

REFERENCES

The references comprise a good range of topic areas and years of publication.

6. PLOS authors have the option to publish the peer review history of their article (what does this mean? ). If published, this will include your full peer review and any attached files.

**Do you want your identity to be public for this peer review?** For information about this choice, including consent withdrawal, please see our Privacy Policy .

Reviewer #1: No

Reviewer #2: **Yes: ** Dr Mary Carter

---

## [Decision Letter · Decision Letter 1]

12 Mar 2025

PGPH-D-24-02343R1

Can audit and feedback improve health service readiness and delivery outcomes in a low-resource setting? Effectiveness results of the ideas strategy from central Mozambique

Dear Dr. Dinis,

Thank you for submitting your manuscript to PLOS Global Public Health. After careful consideration, we feel that it has merit but does not fully meet PLOS Global Public Health’s publication criteria as it currently stands. Therefore, we invite you to submit a revised version of the manuscript that addresses the points raised during the review process.

We look forward to receiving your revised manuscript.

Kind regards,

Annesha Sil, Ph.D.

Staff Editor

PLOS 

Journal Requirements:

Additional Editor Comments (if provided):

Reviewers' comments:

Reviewer's Responses to Questions

**Comments to the Author**

1. If the authors have adequately addressed your comments raised in a previous round of review and you feel that this manuscript is now acceptable for publication, you may indicate that here to bypass the “Comments to the Author” section, enter your conflict of interest statement in the “Confidential to Editor” section, and submit your "Accept" recommendation.

Reviewer #2: All comments have been addressed

Reviewer #3: (No Response)

2. Does this manuscript meet PLOS Global Public Health’s publication criteria ? Is the manuscript technically sound, and do the data support the conclusions? The manuscript must describe methodologically and ethically rigorous research with conclusions that are appropriately drawn based on the data presented.

Reviewer #2: Yes

Reviewer #3: Yes

3. Has the statistical analysis been performed appropriately and rigorously?

Reviewer #2: I don't know

Reviewer #3: Yes

4. Have the authors made all data underlying the findings in their manuscript fully available (please refer to the Data Availability Statement at the start of the manuscript PDF file)?

Reviewer #2: Yes

Reviewer #3: Yes

5. Is the manuscript presented in an intelligible fashion and written in standard English?

Reviewer #2: Yes

Reviewer #3: Yes

6. Review Comments to the Author

Reviewer #2: No further comments.

Reviewer #3: Review comments have been attached - they could not get copied onto this space

7. PLOS authors have the option to publish the peer review history of their article (what does this mean? ). If published, this will include your full peer review and any attached files.

**Do you want your identity to be public for this peer review?** For information about this choice, including consent withdrawal, please see our Privacy Policy .

Reviewer #2: **Yes: ** Dr Mary Carter

Reviewer #3: No

---

## [Decision Letter · Decision Letter 2]

11 Apr 2025

Can audit and feedback improve health service readiness and delivery outcomes in a low-resource setting? Effectiveness results of the ideas strategy from central Mozambique

PGPH-D-24-02343R2

Dear Dr. Dinis,

We are pleased to inform you that your manuscript 'Can audit and feedback improve health service readiness and delivery outcomes in a low-resource setting? Effectiveness results of the ideas strategy from central Mozambique' has been provisionally accepted for publication in PLOS Global Public Health.

Best regards,

Zena Nyakoojo

Associate Editorial Director 

Reviewer Comments (if any, and for reference):

Reviewer's Responses to Questions

**Comments to the Author**

1. If the authors have adequately addressed your comments raised in a previous round of review and you feel that this manuscript is now acceptable for publication, you may indicate that here to bypass the “Comments to the Author” section, enter your conflict of interest statement in the “Confidential to Editor” section, and submit your "Accept" recommendation.

Reviewer #2: (No Response)

2. Does this manuscript meet PLOS Global Public Health’s publication criteria ? Is the manuscript technically sound, and do the data support the conclusions? The manuscript must describe methodologically and ethically rigorous research with conclusions that are appropriately drawn based on the data presented.

Reviewer #2: Yes

3. Has the statistical analysis been performed appropriately and rigorously?

Reviewer #2: I don't know

4. Have the authors made all data underlying the findings in their manuscript fully available (please refer to the Data Availability Statement at the start of the manuscript PDF file)?

Reviewer #2: Yes

5. Is the manuscript presented in an intelligible fashion and written in standard English?

Reviewer #2: Yes

6. Review Comments to the Author

Reviewer #2: Tracked changes on pp 10, 15, 19, 21 and 22 improve the clarity of the manuscript. In the Conclusion on p24, the authors should consider changing 'enhance the complexity of implementing the intervention in this context...' to 'increase the complexity of implementing the intervention in this context...'

7. PLOS authors have the option to publish the peer review history of their article (what does this mean? ). If published, this will include your full peer review and any attached files.

**Do you want your identity to be public for this peer review?** For information about this choice, including consent withdrawal, please see our Privacy Policy .

Reviewer #2: **Yes: ** Dr Mary Carter
